# Bayesian modelling of high-throughput sequencing assays with malacoda

**Andrew R. Ghazi**[1], **Xianguo Kong**[2], **Ed S. Chen**[3], **Leonard C. Edelstein**[2], **Chad A. Shaw**[3]*

**1** Quantitative and Computational Biosciences, Baylor College of Medicine, Houston, Texas, United States of America, **2** Cardeza Foundation for Hematologic Research, Thomas Jefferson University, Philadelphia, Pennsylvania, United States of America, **3** Molecular and Human Genetics, Baylor College of Medicine, Houston, Texas, United States of America

\* cashaw@bcm.edu

**Data Availability Statement:** All relevant data are within the manuscript and its Supporting Information files.

**Funding:** CS, LE - R01HL128234, National Institutes of Health, https://www.nih.gov/ The

## Abstract

NGS studies have uncovered an ever-growing catalog of human variation while leaving an enormous gap between observed variation and experimental characterization of variant function. High-throughput screens powered by NGS have greatly increased the rate of variant functionalization, but the development of comprehensive statistical methods to analyze screen data has lagged. In the massively parallel reporter assay (MPRA), short barcodes are counted by sequencing DNA libraries transfected into cells and the cell's output RNA in order to simultaneously measure the shifts in transcription induced by thousands of genetic variants. These counts present many statistical challenges, including overdispersion, depth dependence, and uncertain DNA concentrations. So far, the statistical methods used have been rudimentary, employing transformations on count level data and disregarding experimental and technical structure while failing to quantify uncertainty in the statistical model. We have developed an extensive framework for the analysis of NGS functionalization screens available as an R package called malacoda (available from github.com/andrewGhazi/malacoda). Our software implements a probabilistic, fully Bayesian model of screen data. The model uses the negative binomial distribution with gamma priors to model sequencing counts while accounting for effects from input library preparation and sequencing depth. The method leverages the high-throughput nature of the assay to estimate the priors empirically. External annotations such as ENCODE data or DeepSea predictions can also be incorporated to obtain more informative priors–a transformative capability for data integration. The package also includes quality control and utility functions, including automated barcode counting and visualization methods. To validate our method, we analyzed several datasets using malacoda and alternative MPRA analysis methods. These data include experiments from the literature, simulated assays, and primary MPRA data. We also used luciferase assays to experimentally validate several hits from our primary data, as well as variants for which the various methods disagree and variants detectable only with the aid of external annotations.

funders had no role in study design, data collection and analysis, decision to publish, or preparation of the manuscript.

**Competing interests:** The authors have declared that no competing interests exist.

## Author summary

Genetic sequencing technology has progressed rapidly in the past two decades. Huge genomic characterization studies have resulted in a massive quantity of background information across the entire genome, including catalogs of observed human variation, gene regulation features, and computational predictions of genomic function. Meanwhile, new types of experiments use the same sequencing technology to simultaneously test the impact of thousands of mutations on gene regulation. While the design of experiments has become increasingly complex, the data analysis methods deployed have remained overly simplistic, often relying on summary measures that discard information. Here we present a statistical framework called *malacoda* for the analysis of massively parallel genomic experiments which is designed to incorporate prior information in an unbiased way. We validate our method by comparing our method to alternatives on simulated and real datasets, by using different types of assays that provide a similar type of information, and by closely inspecting an example experimental result that only our method detected. We also present the method's accompanying software package which provides an end-to-end pipeline with a simple interface for data preparation, analysis, and visualization.

This is a *PLOS Computational Biology* Methods paper.

## Introduction

The advent of next generation sequencing (NGS) has generated an explosion of observed genetic variation in humans. Variants with unclear effects greatly outnumber those with severe impact. For example, the 1000 Genomes Project [1] has estimated that a typical human genome has roughly 150 protein-truncating variants, 11,000 peptide-sequence altering variants, and 500,000 variants falling into known regulatory regions. Simultaneously, genome-wide association studies (GWAS) have found strong statistical associations between thousands of noncoding variants and hundreds of human phenotypes [2,3]. Traditional methods of assessing the regulatory impact of variants are slow and low-throughput: luciferase reporter assays require multiple replications of cloning individual genomic regions, transfection into cells, and measurement of output intensity.

Massively Parallel Reporter Assays (MPRA), overviewed in Fig 1, were developed to assess simultaneously the transcriptional impact of thousands of genetic variants [4]. The simplest form of MPRA uses a carefully designed set of barcoded oligonucleotides containing roughly 150 base pairs of genomic context surrounding variants of interest. There are typically thousands of variants selected using preliminary evidence from GWAS, and there are usually ten to thirty replicates of each allele with unique, inert barcodes. The oligonucleotides are cloned into plasmids, making a complex library that is then transfected into cells. The cells use the library as genetic material and actively transcribe the inserts. Because the barcodes are preserved by transcription, counting the RNA products of each variant construct by re-identifying each barcode in the NGS product provides a direct measure of the transcriptional output of a given genetic variant. By designing the oligonucleotide library to contain multiple barcodes of both the reference and alternate alleles for each variant, one can statistically assess the transcription shift (TS) for each variant. MPRA can thus be used to identify functional driver variants

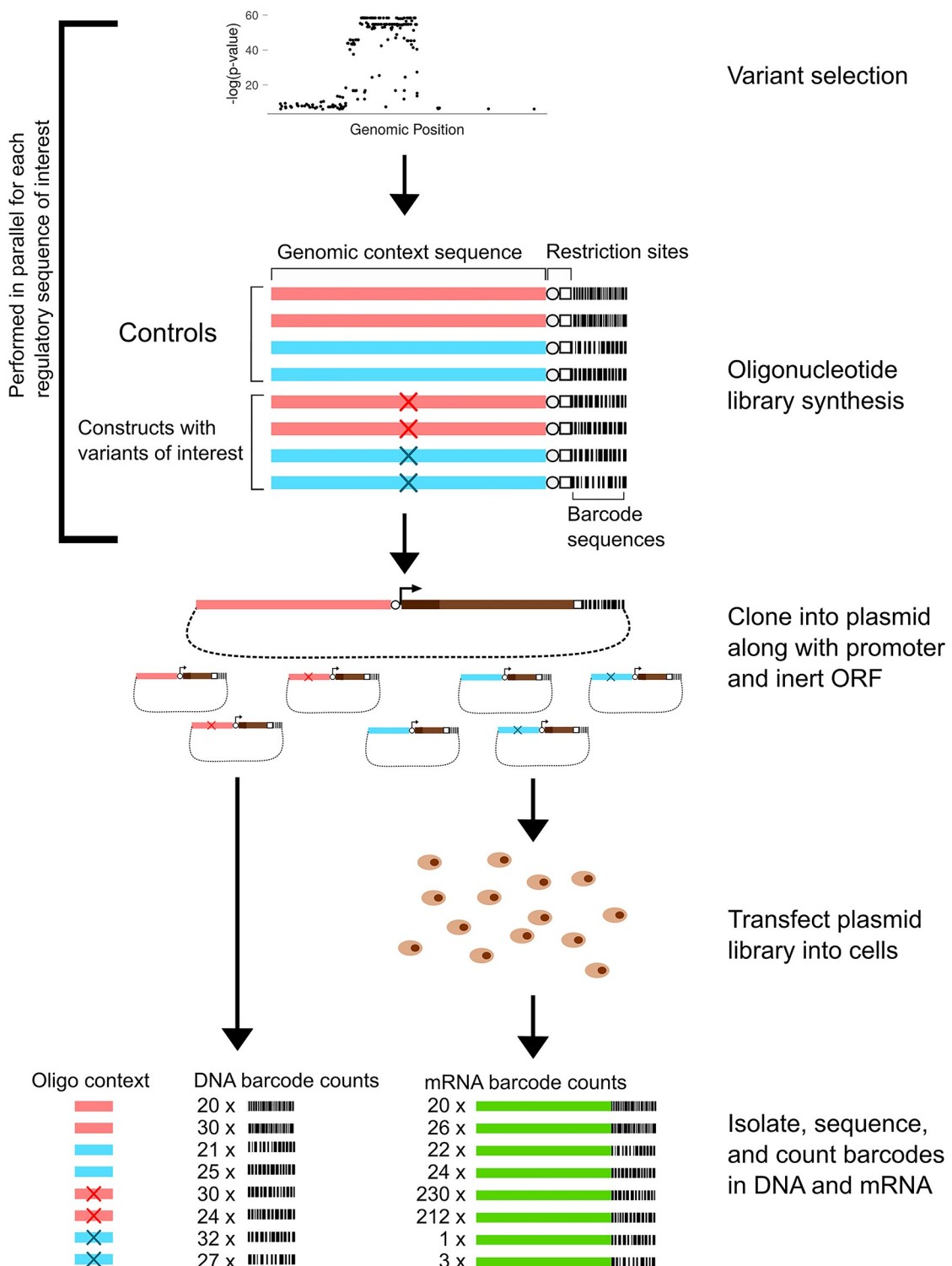

**Fig 1. Diagram of MPRA.** MPRA simultaneously assess the transcription shift of thousands of variants. The diagram shows eight oligonucleotides for two variants (red and blue X's) falling within different regions of genomic context (light red and light blue bars) with two barcodes for each allele of each variant. In practice the complexity and size of the oligonucleotide library is limited only by cost. A typical MPRA has tens to hundreds of thousands of oligonucleotides to assay thousands of variants. The oligonucleotides are cloned into a plasmid library in front of an inert ORF (brown). DNA sequencing of the plasmid library is used to count the input representation of each barcode, then RNA sequencing of the mRNA (green) is used to count the output RNA version of each barcode.

among sets of statistically significant GWAS variants that are difficult to distinguish in observational studies because of linkage disequilibrium.

MPRA have successfully identified many transcriptionally functional variants [5, 6, 7], but the accompanying statistical analyses have been rudimentary. Initial studies focused on the computation of the "activity" for each barcode in each RNA sample. This involves averaging across depth-adjusted counts to compute a normalizing DNA factor for each barcode, then dividing depth-adjusted RNA counts by the DNA factor and taking the log of this ratio. Then a t-test is used to compare the activity measurements for each allele, followed by assay-wide multiple-testing corrections. The key limitations include ignoring systematic variation due to unknown DNA concentrations, compounded data transformation and summarization prior to modelling, and the failure to include the reservoir of prior data and biological knowledge concerning genes and genomic regions. The methods mpralm [8], MPRAscore [9], QuASAR-MPRA[10], and MPRAnalyze [11] are more recent methods, but they all suffer from some combination of common limitations: failure to model variation in input DNA concentrations, aggregation of data across barcodes, sequencing samples without modelling systematic sources of variation, and over-reliance on point estimates of dispersion that cause errors in transcription shift estimates.

Other areas of genomic analysis have generated a wealth of information on genomic structure and function, frequently specific to particular genomic contexts and variants. For example, the ENCODE project [12] provides genome-wide ChIP-seq data on transcription factor binding profiles, histone marks, and DNA accessibility. Computational methods such as DeepSea [13] use machine learning to provide variant-specific predictions on chromatin effects. Genome-wide databases like ENCODE and computational predictors like DeepSea contain real information about variant effects, but a method for incorporating this information into a statistical framework for experimental analysis of variants has not been developed.

We hypothesized that a structured, probabilistic modelling approach to high-throughput NGS screens such as MPRA would yield more accurate estimates of variant function while improving statistical sensitivity and specificity, particularly when incorporating prior information. This approach offers a flexible modelling system that can fit hierarchical model structures of count data while also directly accounting for experimental sources of variation. Our approach would also enable the integration of prior information and account for uncertainty in dispersion parameter estimates. These advantages offer significant improvements in statistical efficiency and provide opportunities for formulating systems-level hypotheses—for example, the impact of specific transcription factors—that are absent from other approaches. Here we present *malacoda*, an end-to-end Bayesian statistical framework that addresses gaps in the prior approaches while providing novel methods for incorporating prior information. The malacoda method focuses on MPRA but also has potential extension to a broad array of NGS-based high-throughput screens. We establish the superior performance of malacoda on MPRA compared to alternatives using simulation studies. We then apply the method to previously published findings to make new biological discoveries that we explore in the paper. We also apply malacoda to primary MPRA studies that we performed. We limit the analysis of our primary data to an examination of the inter-method consistency of effect size estimates in order to emphasize the potential of our statistical method. The barcode counts and cross-method effect size estimates for all of the results are included in S2 Data. To demonstrate the impact of malacoda for biologically relevant discovery, we analyzed previously published data by Ulirsch et al, and we identified the functional variant rs11865131 within the intron of the *NPRL3* gene; we validated this finding by luciferase assay. The results demonstrate that using malacoda we can discover biologically important findings that were missed by prior approaches. We have made the software available as an open source R package on GitHub.

## Methods

### Overview

In malacoda we utilize a negative binomial model for NGS to consider barcode counts with empirically estimated gamma priors, and we explicitly model variation in the input DNA concentrations for each barcode. By default, the method marginally estimates the priors from the maximum likelihood estimates of each variant in the assay; the method also supports informative prior estimation by using external genomic annotations for each variant as weights. This approach enables disparate knowledge sources to inform the results in a principled, data-driven way. The probabilistic model underlying malacoda uses the NGS data directly without transformation, and it accounts for all known sources of experimental variation and uncertainty in model parameters. Finally, the method provides estimate shrinkage as a method for avoiding false positives.

### Description of the statistical model

MPRA data are composed of the counts of the barcoded DNA input from sequencing the plasmid library and the counts of the barcoded RNA outputs from sequencing the RNA content extracted from passaged cells. The DNA counts vary according to the sequencing depth of the sample as well as due to the inherent noise in library preparation. The RNA measurements also vary according to sequencing depth, but they are also affected by the DNA input concentration and the inherent transcription rate of their associated region of genomic context. Fig 2A shows a subset of a typical MPRA dataset, with two barcodes of each allele for two variants and several columns of counts. We find that typically MPRA are performed with four to six RNA sequencing replicates and a smaller number of DNA replicate samples. Fig 2B shows a simplified Kruschke diagram of the model underlying malacoda, using the mean-dispersion parameterization of the negative binomial. More explicitly,

$$\mu_{\mathrm{DNA}_{bc}} \sim \mathrm{Gamma}(\alpha_{\mu_{\mathrm{DNA}}}, \beta_{\mu_{\mathrm{DNA}}})$$

$$\mu_{\mathrm{allele}} \sim \mathrm{Gamma}(\alpha_{\mu_{\mathrm{RNA}}}, \beta_{\mu_{\mathrm{RNA}}})$$

$$\phi_{\mathrm{DNA}} \sim \mathrm{Gamma}(\alpha_{\phi_{\mathrm{DNA}}}, \beta_{\phi_{\mathrm{DNA}}})$$

$$\phi_{\mathrm{allele}} \sim \mathrm{Gamma}(\alpha_{\phi_{\mathrm{RNA}}}, \beta_{\phi_{\mathrm{RNA}}})$$

$$\mathrm{Counts}_{\mathrm{DNA}_{s,bc}} \sim \mathrm{NegBin}(\mathrm{mean} = d_s \times \mu_{\mathrm{DNA}_{bc}}, \mathrm{dispersion} = \phi_{\mathrm{DNA}})$$

$$\mathrm{Counts}_{\mathrm{RNA}_{s,bc}} \sim \mathrm{NegBin}(\mathrm{mean} = d_s \times \mu_{\mathrm{DNA}_{bc}} \times \mu_{\mathrm{allele}}, \mathrm{dispersion} = \phi_{\mathrm{allele}})$$

Where $d_s$ indicates the depth of a particular sequencing sample, $\mu_{\mathrm{DNA},bc}$ indicates the unknown concentration of a particular barcode in the plasmid library, and $\mu_{\mathrm{allele}}$ indicates the effect of the genomic context of a given allele of a given variant. Parameters indexed by "bc" are vectors with an element for each barcode while those with the "allele" subscript contain two elements for the reference and alternate alleles. The shape $\alpha$ and rate $\beta$ parameters of the Gamma priors are estimated empirically. Note that the mean of each negative binomial used to model a particular count observation is directly proportional to the sequencing depth of the sample from which that count observation arose. A more finely detailed walkthrough of the model and its implementation are available in section 1 of S1 Appendix.

A

| variant_id | allele | barcode | MPRA_DNA1 | MPRA_DNA2 | MPRA_RNA1 | MPRA_RNA2 | MPRA_RNA3 |
|---|---|---|---|---|---|---|---|
| 7_79758455_C_T_CD36 | ref | GCCATAAGCAGTCT | 473 | 788 | 3329 | 8337 | 5106 |
| | | TTACGAATAGTGCG | 362 | 549 | 3571 | 7342 | 4259 |
| | alt | TAGCTGTTCCTGAC | 1807 | 2887 | 1788 | 4422 | 3166 |
| | | ATGCCGTTGCGATT | 48 | 48 | 40 | 0 | 48 |
| rs11749731 | ref | AACCGTCGCGTAGT | 543 | 868 | 248 | 759 | 489 |
| | | CACGCAATGTCTTA | 173 | 246 | 93 | 89 | 75 |
| | alt | CTTCGTACTATTCC | 412 | 638 | 370 | 685 | 707 |
| | | AGGACGCAATACAA | 284 | 569 | 520 | 1107 | 1090 |
| rs2236053 | ref | CTACCGCGTCACTA | 457 | 660 | 1616 | 3875 | 3164 |
| | | AGGTGACTTGTAGG | 69 | 123 | 314 | 366 | 365 |
| | alt | ATCTGTCGCGCTAT | 165 | 248 | 540 | 998 | 593 |
| | | TAGCGTGTACTTCA | 1122 | 1708 | 3819 | 8397 | 6181 |

B

$\alpha_\mu$ , $\beta_\mu$

gamma

$\alpha_\phi$ , $\beta_\phi$

gamma

Estimated from other alleles within the assay
+
{annotation based similarity}

$\mu$ , $\phi$

neg. binomial

Evaluate joint posterior

MPRA Counts

Transcription Shift Posterior

C

Marginal prior

Grouped prior — Informative grouping

Conditionally weighted prior — Informative predictor

**Fig 2. MPRA data and malacoda priors.** A) The table shows a subset of our primary MPRA data. The highlighted cell containing 759 barcode counts is influenced both by the sequencing depth of its sample (blue column) and the unknown input DNA concentration of its barcode (red row). B) A simplified Kruschke diagram of the generative model underlying malacoda. After evaluating the joint posterior on all model parameters, a 95% posterior interval on a variant's transcription shift (shaded area) may be used for a binary decision between "functional" or "non-functional". This example TS posterior shows a negative shift that excludes zero, meaning the variant in question would be called as "functional". C) A conceptual diagram demonstrating three prior types available in the malacoda framework. The marginal prior (left) weights all variants in the assay equally, while the grouped and conditional priors utilize informative annotations as weights in the prior estimation process.

The negative binomial distribution is a natural choice for modelling NGS count data given its ability to accurately fit overdispersed observations frequently seen in sequencing data [14]. Briefly, the observed dispersion in NGS count data usually exceeds that expected from simpler binomial or Poisson models. We chose gamma distributions as priors for several reasons. They have the appropriate $[0, \infty)$ support, and for a non-negative random variable whose expectation and expected log exist, they are the maximum entropy distribution. Additionally, they are characterized by two parameters, which gives the prior estimation process enough flexibility to accurately fit the observed population of negative binomial estimates. Probabilistic modelling of the dispersion parameters is key as demonstrated by simulation in S2 Appendix. Allocating probability across a distribution of dispersion parameter values impacts the inference on the other parameters in the model, specifically the allele-level effects that the assay aims to evaluate. The practice of modelling dispersion parameters probabilistically helps avoid pitfalls found in methods that utilize point estimates of dispersion. This barcode-level count data model that quantifies the uncertainty on the dispersion parameters is a central contribution of the malacoda method.

After computing the joint posterior on all model parameters, the posterior on transcription shift is computed as a generated quantity by taking the difference between log of $\mu_{allele}$ for the alternate and reference alleles. We then compute the narrowest interval containing 95% of the posterior on TS (the highest density interval (HDI)) for each variant. The 95% HDI is used to make binary calls on whether a variant is functional or non-functional: if the interval excludes zero as a credible value, the variant is labelled as "functional". We note here that 95% is an arbitrary threshold based on statistical convention and common values on the trade-off between sensitivity-specificity. Other common cutoffs such as 80% or 99% may be used. An optional "region of practical equivalence" may also be defined on a per-assay basis when there is particular interest in rejecting a null region of transcription shift values around zero [15].

## Empirical priors

The gamma priors are fit empirically using maximum likelihood estimation. Specifically, each variant-level model is fit first by maximizing the likelihood component of the malacoda model, then empirical gamma distributions are fit to those estimates for the means and dispersions of the DNA, reference RNA, and alternate RNA. This approach offers several benefits. First, it leverages the high-throughput nature of the assay. The full dataset of thousands of variants determines the prior, so the contribution from each individual variant is small. Secondly, it constrains the prior to be reasonable in the context of a given assay. Specific circumstances regarding library preparation, sequencer properties, cell culture conditions, and other unknown factors will cause the underlying statistical properties of each MPRA to be unique. A less informed, general-purpose prior, such as Gamma(shape = 0.001, rate = 0.001), would assign a considerable amount of probability density to unreasonable regions of parameter space. Empirical estimation ensures that the priors capture the reasonable range of values for each parameter while avoiding putting unwarranted density on extreme values [16]. Finally, by sharing information between variants, empirical priors provide estimate shrinkage. The prior effectively regularizes all parameter estimates, a behavior which is important in multi-

parameter models with relatively little data per parameter. This regularizing effect acts as an alternative to *post hoc* multiple testing correction: rather than widening the confidence interval on the estimate of the transcription shift, an empirical prior shrinks the estimate of transcription shift towards the global average while leaving the width of the interval intact. This data-driven approach acts as a natural safeguard against the risk of false positives found in multiple testing scenarios while simultaneously moderating the reported effect sizes of variants that display extreme behavior by chance. The regularization effect of the empirical prior is demonstrated in section 6 of S3 Appendix.

In order to incorporate external knowledge, the malacoda method also allows users to provide informative annotations to supplement the analysis. Fig 2C contrasts the marginal prior (left) with two prior types that make use of external annotations. These priors use the information in the annotations by employing the principle that similarly annotated variants should perform similarly in the assay. When the annotations are simply a set of descriptive categories (for example predictions of likely benign, uncertain, or likely functional), the grouped prior (2C, center) simply fits a prior distribution within each subset. When the annotations are continuous values, the conditionally weighted (2C, right) prior employs an adaptive kernel smoothing process to estimate the prior. To estimate the prior for a single variant, it initializes a t-distribution kernel centered at the annotation of the variant in question, then gradually widens this kernel until the *n*-th most highly weighted variant (where *n* is a configurable tuning parameter defaulting to 100) has a weight of at least one percent of that of the most influential variant. This ensures that the weights used to estimate the conditional prior are not dominated by the nearest neighbor in annotation space. While the diagram in Fig 2C shows this for only a single informative annotation on the horizontal axis, the software allows for an arbitrary number of continuous predictors to be used.

## Simulation and validation studies

We took several approaches to validate and compare the malacoda method with alternatives. First, we simulated MPRA data across a realistic grid of parameters governing the fraction of truly functional variants, the number of variants in the assay, and the number of barcodes per allele. These simulations also modelled distinct sequencing samples, realistic variation in sequencing depth, and barcode failure during library preparation. We then compared malacoda to alternative methods including the t-test, mpralm, MPRAscore, QuASAR-MPRA, and MPRAnalyze. Across these simulations we compared performance metrics including area under the receiver operating characteristic curve (AUC), area under the precision-recall curve (AUPR) and estimate accuracy. The code used to generate these simulations is provided in sections 2 and 3 of S3 Appendix. Secondly, we applied malacoda and alternative methods to real MPRA data from the Ulirsch dataset [5], using inter-method consensus as a performance metric. We repeated this using our own primary MPRA data from an assay performed in K562 cells inspecting 2666 variants related to platelet function. This assay utilized oligonucleotides with 150bp of genomic context and inert 14bp barcodes. The barcode counts from this assay are presented in S2 Data. In both cases we ran malacoda using both a marginal prior and a conditional prior informed by DeepSea predictions for DNase hypersensitivity in the relevant cell-type. Finally, we tested a subset of variants with luciferase reporter assays to assess consistency with MPRA estimates of variant function.

## Computational methods and software

Our method is available as an R package from github.com/andrewGhazi/malacoda. The package includes detailed installation instructions, extensive help documentation, an analysis

walkthrough vignette, and implementations of traditional activity-based analysis methods. The statistical models are fit with Stan [17], which allows us to perform a fast first pass fit with Automatic Differentiation Variational Inference [18] and, if a narrow 80% posterior interval on TS excludes zero, to perform a final Markov Chain Monte Carlo (MCMC) fit with Stan's No-U-Turn Sampler. This presents an effective balance between the speed of approximate variational inference and asymptotically exact estimation of parameters via MCMC for functional variants. By default, each variant is first checked with a variational first pass. Then, if the variant passes the posterior interval check, MCMC is performed with 4 chains using 200 warmup and 500 post-warmup samples per chain for a total of 2000 posterior samples. These default settings can refine the limits of the TS posterior interval with satisfactory precision within a short run time. While the adaptive Hamiltonian Monte Carlo provided by Stan can efficiently explore high-dimensional posteriors, any MCMC-based method has Monte Carlo error that makes estimate precision difficult in borderline situations—an additional digit of estimate precision requires 100 times as many MCMC samples. When using the 95% posterior interval to make a binary classification of functional or non-functional, variants on the borderline can require a large number of posterior samples to precisely refine the limits of the interval that is used to classify a variant as functional or non-functional. By default, malacoda checks to see if either edge of the 95% interval is close to zero, and if necessary, lengthens the MCMC chains in order to provide better precision in this scenario. A full walkthrough of the computational methods is provided in section 2 of S1 Appendix.

Our package also includes data processing functionality to extract barcodes from reads, filter barcodes by quality, and count barcodes from a set of FASTQ files through an application of the FASTX-Toolkit [19]. Through an interface with the FreeBarcodes package [20], the package can also decode sequencing errors in the barcodes of an assay that has been designed using our previous work, mpradesigntools [21]. In our experience this typically recaptures about 5% additional data with no additional cost beyond a line of code during the assay design process. The package also contains plotting functionality to help visualize the results of analyses.

## Experimental methods

In order to collect experimental measurements of the transcriptional impact of variants through means other than MPRA, we performed luciferase reporter assays on seventeen variants. Four were among the strongest signals detected in our MPRA, six were variants from our MPRA where the statistical methods disagreed, and seven were variants from the Ulirsch dataset [5] where the malacoda marginal and DeepSea-based [13] conditional prior model fits disagreed.

150-200bp genomic DNA sequences flanking the variants were amplified by PCR using K562 lymphoblast (ATCC) genomic DNA as template, then cloned into PGL4.28 minimum promoter luciferase reporter vector (Promega) at NheI and HindIII sites. Counterpart SNP variants were generated by site-directed mutagenesis. All the constructs were validated by DNA sequencing. 3μg plasmid preparations were co-transfected with 0.5μg β-gal plasmid into $1x10^6$ of K562 cells with Lipofectamine 2000 based on manufacturer's instructions. Each assay was repeated with 3 independent plasmid preparations. 24 hours post transfection, luciferase and β-gal were measured. Luciferase units were then normalized to β-gal values. These results are available in S1 Data.

## Results

### Simulation studies

We evaluated our simulation results in three ways. First, we examined the accuracy of transcription shift estimates. Fig 3A shows the results of analyzing one simulated dataset, with the

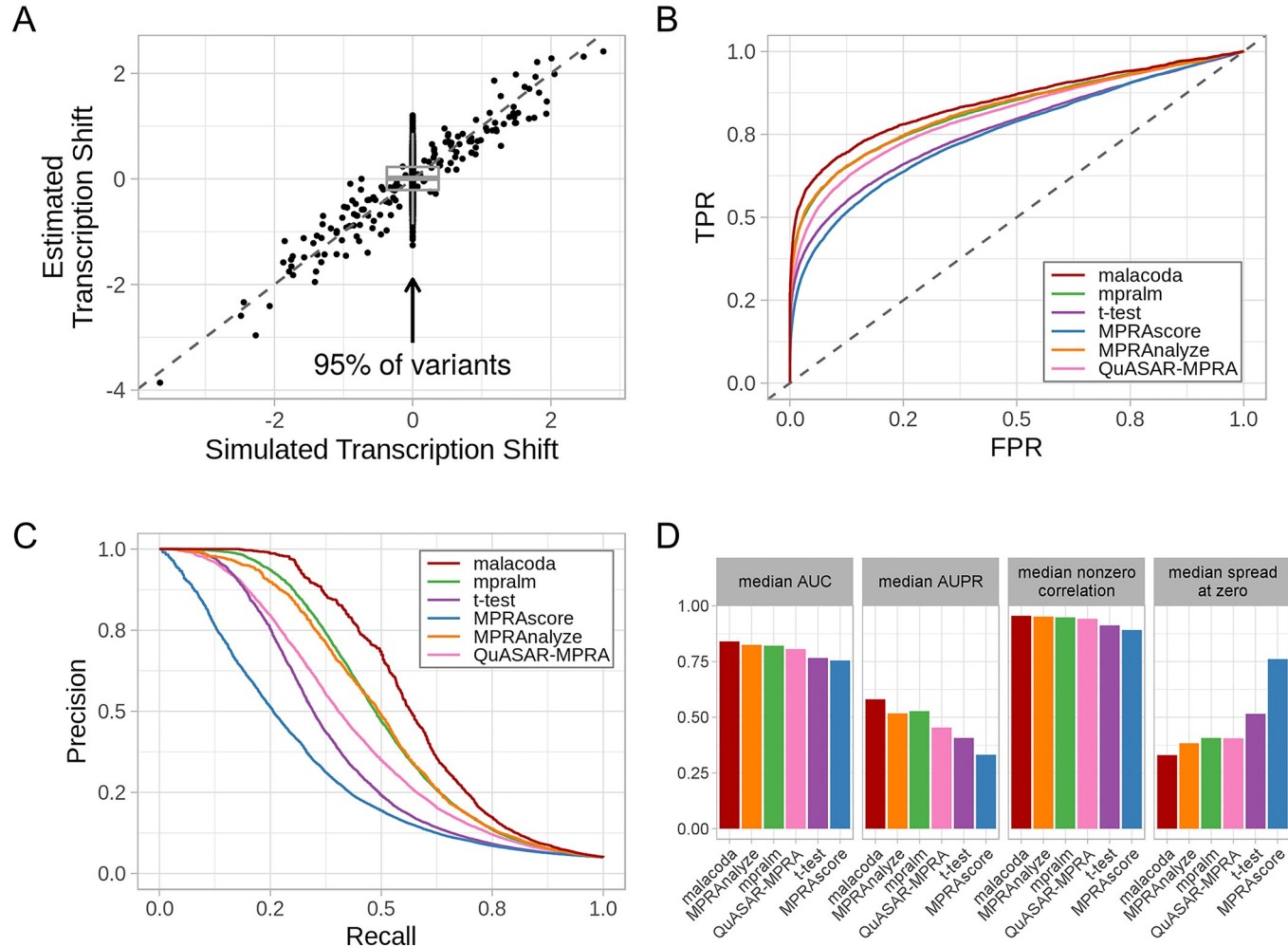

**Fig 3. Simulation results.** A) The figure compares the TS values used to generate simulated data to TS estimates. Simulated MPRA assays use a varying fraction of variants that are truly non-functional (center). B) The average ROC curves used to assess the classification performance of each method across simulations with 3000 variants, 5% truly functional variants, and 10 barcodes per allele. The methods shown are malacoda (red), MPRAnalyze (orange), mpralm (green), QuASAR-MPRA (pink), MPRAscore (blue), and the t-test (purple) C) The average precision-recall curve for the same set of simulations D) Median performance metrics across multiple simulations under the same conditions as B.

true value of the simulation's transcription shift plotted on the x-axis, with the model estimates on the y-axis. For each fit of each simulation using each analysis method, we analyzed accuracy using two metrics: standard deviation of estimates for truly non-functional variants at zero (vertical width of the grey boxplot, lower is better) and correlation with the true values for simulated functional variants with nonzero effects (off-center points, higher is better).

Second, we also computed area under the receiver operating characteristic curve (AUC) and area under the precision-recall curve (AUPR) in order to characterize the binary classification performance of each method. Bayesian methods such as malacoda explicitly do not consider a null hypothesis and therefore do not output p-values. In order to create an analogous quantity needed to compute the AUC and AUPR, we instead computed one minus the minimum HDI width necessary to include zero as a credible transcription shift value to distinguish true and false positives. This process is presented in detail in section 4.1 of S3 Appendix. Fig 3B shows the ROC curves by method averaged over simulated assays with ten barcodes per allele, 5% truly functional variants, and 3000 variants. Fig 3C shows the precision-recall curves

for the same simulations. Fig 3D shows that across all simulations with these characteristics, malacoda consistently showed the highest median AUC and AUPR, the highest correlation with the truth for functional variants, and the lowest standard deviation of estimates of truly non-functional variants. The last metric, "spread at zero", particularly emphasizes the regularization effect, showing that while malacoda tends to produce the most accurate effects for functional variants, it can simultaneously provide the smallest estimates for truly non-functional variants. Other combinations of simulation parameters are shown in section 5 of S3 Appendix, displaying similar patterns.

In order to examine the performance of malacoda on real data, we applied the various methods to both the Ulirsch data [5] and to our own primary dataset. Unlike the case with simulations, the underlying true transcription shift values are not known. However, inter-method consensus can serve as a performance metric. Methods that utilize varying model structure will tend to make errors in different ways, so methods that consistently perform well will show higher correlation with alternatives than the correlations between the methods that perform poorly. Indeed, Fig 4 shows that the other methods tend to correlate with malacoda better than each other. This occurs despite the expected non-linear relationship between regularized and unregularized models (i.e. between malacoda and the other alternatives). The fits based on malacoda's marginal and conditional priors (first and second rows/columns) in both panels of Fig 4 tend to correlate strongly because of the identical model structure paired with large spread of DeepSea predictions used in the prior estimation process. The conditional prior fit only deviates significantly from the marginal prior fit for variants with high DeepSea predictions.

## Biological results

The variants we tested with luciferase reporter assays were predominantly chosen from the set where malacoda's marginal and conditional fits disagreed on functionality, not those variants showing the strongest effects. These discordant variants tended to have small effects and the noise between replicates tended to be comparable to the mean intensity ratio. Therefore, the number of variants tested was not enough to overcome the noise inherent to light intensity-based measurements and provide conclusive results on the accuracy of the various MPRA analysis methods. While we were able to recapitulate the transcriptional functionality of several variants, we did not have enough data to clearly demonstrate that any of the MPRA analysis methods outperform the others in terms of correlation with luciferase results. Nonetheless, S2 Fig shows that the various methods are consistent with MPRA-based estimates for variants with large shifts, providing further evidence that MPRA results are biologically realistic.

We closely inspected a particular biological discovery to demonstrate malacoda's ability to identify low-signal variants. One of the functional variants we identified with malacoda using the DeepSea-based conditional prior in the Ulirsch dataset [5] is rs11865131; this variant is identified by malacoda but not by any of the other methods after multiple testing corrections or with the marginal prior. The conditional prior is compared to the marginal prior in S1 Fig. We validated this variant is functional by luciferase assay in K562 cells with the results shown in Fig 5. The variant rs11865131 is in an intron within the *NPRL3* gene which encodes the Natriuretic Peptide Receptor Like 3 protein. *NPRL3* is part of the GTP-ase activating protein activity toward Rags [22] (GATOR1) complex. The GATOR1 complex inhibits mammalian target of rapamycin (*MTOR*) by inhibiting *RRAGA* function (reviewed in [22] *MTOR* signaling has been implicated in platelet aggregation and spreading in addition to aging associated venous thrombosis [23, 24]. Analysis of the rs11865131 locus with HaploReg [25] indicates that it colocalizes with ENCODE ChIP-Seq peaks for 36 bound proteins (predominantly

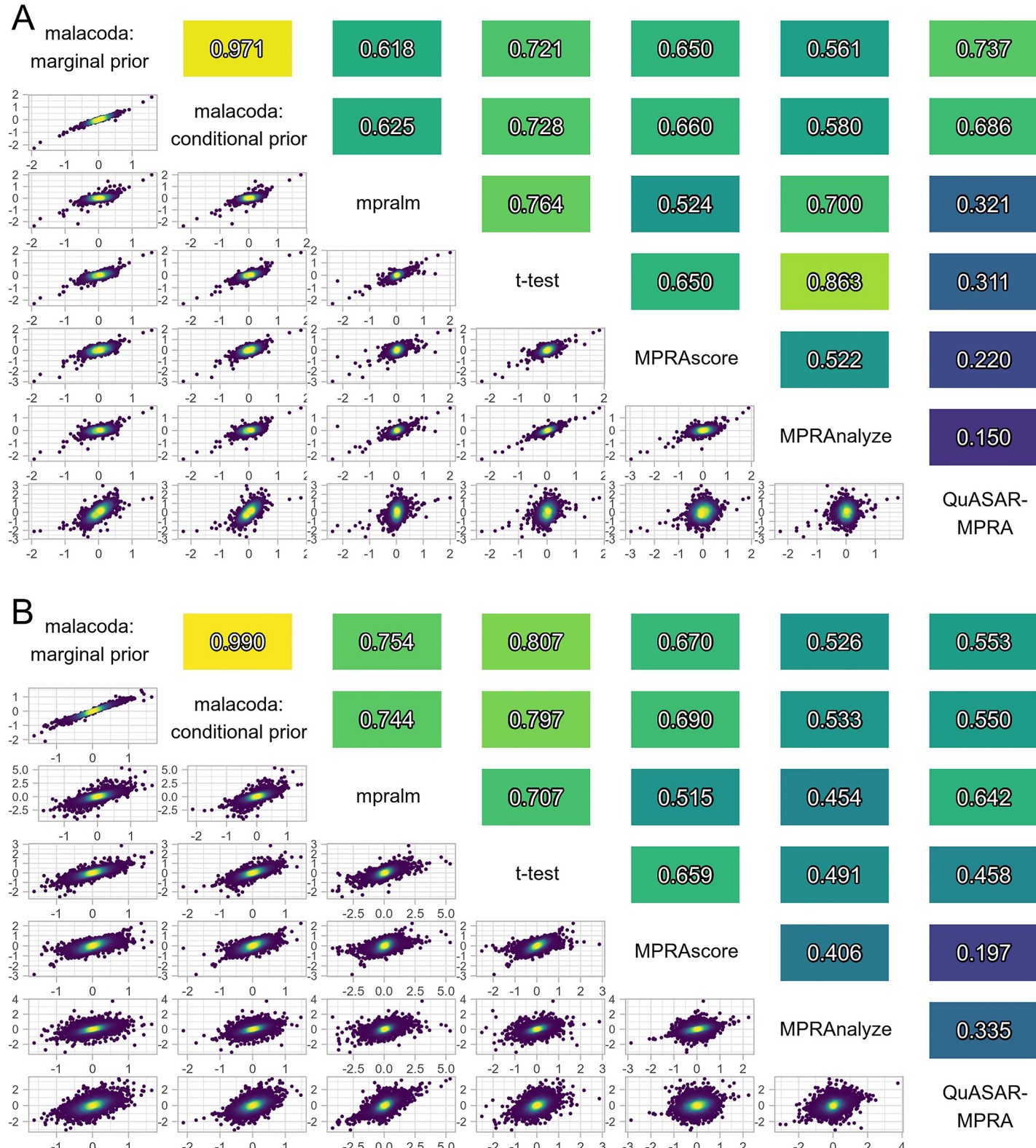

**Fig 4. Inter-method consensus.** A) A pairwise plot of TS estimate comparisons between methods in our primary MPRA dataset, showing that alternative methods generally agree with malacoda more than each other. Shaded values above the diagonal show the correlation values for the corresponding plot below the diagonal. Color below the diagonal indicates local density of points in over-plotted regions. B) A pairwise plot of TS estimates using both the marginal and DeepSea-based malacoda priors in the Ulirsch dataset, showing a similar outcome.

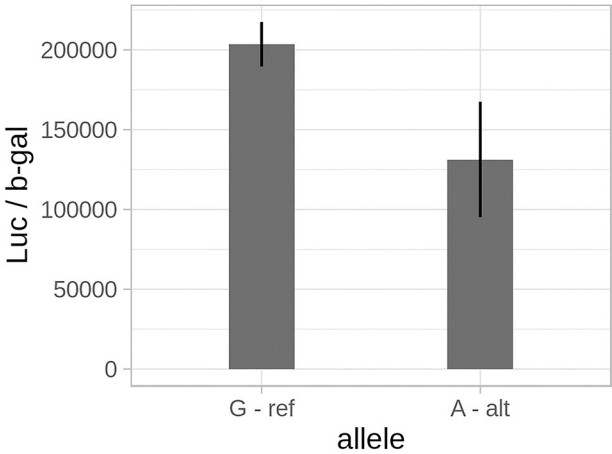

**Fig 5. Luciferase validation results.** A bar plot showing the difference in normalized luciferase intensity for both alleles of rs11865131 (p = 0.032). Black error bars indicate +/- one standard deviation.

transcription factors) in K562 erytholeukemia cells as well as containing enhancer histone epigenetic marks. Furthermore, this variant lies roughly one thousand base pairs away from the nearest exon-intron boundary, suggesting that it is unlikely to alter splicing of the NPRL3 transcript. Together, these data indicate that this is likely an important regulatory region. In addition to the heterologous K562 cell line, data from cultured megakaryocytes indicates that rs11865131 lies within *RUNX1* and *SCL* ChIP-Seq peaks, two well-studied megakaryopoietic transcription factors [26]. This agrees with our data that platelet *NPRL3* mRNA is positively associated with platelet count in healthy humans [27, 28]. These data indicate that malacoda has identified a likely important regulatory region for megakaryocytes and platelets that was missed by other MPRA analysis methods.

MCMC can be computationally expensive, so we measured the run times in our study. The computational performance was first evaluated using the default settings of the malacoda package which are set to strike a balance between speed and precision for exploratory analysis. These settings include the variational first pass, 200 warmup samples, four chains yielding a total of 2000 posterior samples, and adaptively increased chain lengths. This initial analysis run of 8251 variants from the Ulirsch dataset took 29 minutes when parallelized across 18 threads on two Intel Xeon X5675 3.07GHz processors. We compared this to a highly precise analysis run on the same dataset with no variational first pass and excessively long 50,000 iteration MCMC chains for all variants, which took fifteen hours with the same number of cores on the same processors. The correlation between posterior mean TS between these two runs was 0.981 for non-functional variants and 0.99996 for functional variants. This result, together with the MCMC diagnostics shown in section 2.4.2 of S1 Appendix, demonstrates that the sampler used by our software is able to produce accurate estimates in a relatively short amount of time. Details of the computational methodology and results demonstrating convergence are presented in section 2.4 of S1 Appendix.

## Discussion

We developed a fully Bayesian framework for the analysis of NGS high-throughput screens with particular focus on MPRA studies. The method, called malacoda, is an advance in statistical and computational science that probabilistically incorporates all known sources of

variation for these high throughput NGS screens. The method does a better job of identifying true positives in simulated data and performs well in empirical studies. We also showed that the method identified a previously overlooked functional variant in the *NPRL3* gene that has confirmatory evidence from a variety of other studies. Particular advantages of the method are accurate estimation of variant effects, the treatment of the dispersion parameter in both estimation and inference, and the potential to incorporate informative prior information.

The functional discovery of the variant rs11865131 represents a demonstration of the power of the malacoda method to identify biologically important results missed by alternative methods. This variant lies in an intronic region of the gene *NPRL3*, meaning approaches focused on alterations to the gene's protein code would overlook this regulatory variant. Multiple lines of evidence point to the biological relevance of this variant, including epigenetic and transcription factor binding data as well as evidence of association with platelet count in healthy humans.

There are downsides to our method. First, Bayesian methods that estimate a joint posterior on many parameters by MCMC are significantly slower than optimization-based approaches. We took several approaches to mitigate this, utilizing Stan's No-U-Turn Sampler and including options for first pass variational approximations, adaptive MCMC length, and parallelization. Together these features enable relatively fast model fitting. Second, our method does not account for uncertainty in our empirical prior estimation procedure [16]. Our R package includes a fully hierarchical model that adds an additional layer of hyperparameters in order to probabilistically model the gamma prior parameters at the same time as all of the variant-level parameters. This provides a joint posterior that models an entire MPRA dataset with a single MCMC fit. However, this model, featuring hundreds of thousands of parameters when used in the context of a typical MPRA, is presently too complex to fit in practice and was not used for the results presented in this work. Finally, our work is limited to MPRA performed in K562 cells, however there is nothing cell-type specific about the malacoda model. Our method can be used in MPRA performed in alternative cell-types so long as they follow the experimental structure outlined in the Methods section.

It is worthwhile to discuss the most effective ways to utilize external annotations to estimate informative empirical priors. We encourage users to utilize information that was originally used in the assay's variant selection process. For example, assays designed around inspecting specific transcription factors with varying biological context may want to use the targeted transcription factor as the group identifier in a grouped prior as in Fig 2C. Using information independent of the original design can also be helpful, as we have demonstrated through the use of a conditional prior based on DeepSea's K562 DNase hypersensitivity predictions which helped to refine the inference on a low-signal variant, rs11865131. The malacoda package can utilize an arbitrary number of continuous annotations, so any set of relevant, independent annotations may be used. As long as the principle of "similarly annotated variants have similar outcomes in the assay" holds, using informative annotations can help refine the analysis. Nonetheless, it is difficult to accurately predict the transcription shift of a single variant *a priori*. Conditional priors that make strong predictions of functionality should be treated with caution. We encourage the users to utilize the prior visualization functionality included in the package to contrast annotation-based priors against a marginal prior. Future advances in machine learning models for predictive variant annotation will likely improve the performance of the informative empirical priors.

It is desirable to identify an orthogonal gold-standard dataset to differentiate the accuracy of MPRA analysis approaches. Such an analysis would define an independent score of functionality for all variants, and then hits and non-hits from each MPRA analysis method could be compared for their concordance or correlation with this independent score. We attempted

such an analysis using the Ulirsch dataset, ENCODE K562 bound protein levels, and DeepSea DNase hypersensitivity annotations. Unfortunately these analyses were inconclusive, showing no clear difference in annotation scores between analysis methods. There are at least two possible explanations for this difficulty. First, the noise present in both the MPRA and annotation data lowers the power to differentiate the methods. Secondly, there is misalignment between MPRA functionality and differential scoring in the annotation data. Both of these factors likely contribute to the negative result. We would postulate that if there were an idealized dataset showing high correspondence in variants that are potentially functionalizable by MPRA and simultaneously differentially scored in the orthogonal annotation data, then this hypothetical data could be used to compare the efficacy of the various MPRA analysis methods. At present, we know of no data source that would meet these requirements. While this limits our ability to quantify the performance of MPRA analysis methods, it speaks to the value of MPRA themselves. MPRA produce a unique biological signal that cannot be easily measured by other types of experiments or data.

The statistical method and validation work presented in this article present many future directions in the statistical analysis of high-throughput sequencing assays. This article has focused primarily on the analysis of "typical" MPRA: two alleles per variant, in a single tissue type, with no other experimental perturbations. However, we have expanded the modelling capabilities of the software package beyond these limitations. Models tailored to more complicated experimental structures, such as arbitrary numbers of alleles per variant, multiple tissue types, or cell-culture perturbations, are also included with the package. We also have expanded the model framework included in the package to CRISPR screen modelling. In this CRISPR model, the counts of gRNAs targeting specific genes in survival/dropout screens can make use of an analogous negative binomial structure with similar empirical gamma priors. This opens the path to incorporating gene-level annotations into Bayesian CRISPR screen analysis.

Sophisticated high-throughput assays are a central component to the future of genomics. Therefore, the statistical methods used for these data should be as efficient as possible, accounting for all sources of variation and quantifying the resulting uncertainty. Our software, malacoda, provides an end-to-end framework for the probabilistic analysis of MPRA data. Through our well-documented, easy-to-use R package, users can perform sequencing error correction and data pre-processing before executing a fully Bayesian analysis in as little as two lines of code. The method is capable of taking advantage of informative annotations through an adaptive empirical prior estimation. We hope that this work may act as a stepping stone towards further integrative, probabilistic analysis in the field of high-throughput genomics.

## Supporting information

**S1 Appendix. Model description, fitting, and diagnostics.**
(PDF)

**S2 Appendix. Negative Binomial variance estimation.**
(PDF)

**S3 Appendix. Simulation details and extended results.**
(PDF)

**S1 Data. RData file of luciferase and primary MPRA results.** An RData file that loads two objects: luc_results, a table of the luciferase results, and mpra_results, giving the primary data on MPRA counts for the variants tested with luciferaseF.
(RDATA)

**S2 Data. RData file of estimate comparisons and primary MPRA data.** An RData file that contains three data frames: ulirsch_comparisons, primary_comparisons, and primary_mpra_-data. The first two data frames are the data necessary to produce Fig 4. Each row corresponds to one variant, and each column corresponds to a given analysis method. The values in the table give the transcription shift estimates. The third data frame gives the barcode counts from our primary MPRA dataset with anonymized variant identifiers.
(RDATA)

**S1 Fig. Prior comparison plot for rs11865131.** This figure compares the allelic priors for the RNA activity for both alleles of rs11865131. The blue line shows the marginal prior, the red line the conditional prior based on the DeepSea K562 DNase hypersensitivity prediction. Dotted lines show the prior means. Black tick marks show the RNA count observations adjusted for sequencing depth and DNA input. Because this variant tended to show higher than usual activity in both alleles, both priors shrink the activity considerably. Notably however, the conditional prior shrinks less than the marginal, particularly in the reference allele. The allele-specific difference in shrinkage is what allowed the conditional prior-based analysis to identify this variant as functional.
(TIF)

**S2 Fig. Luciferase versus MPRA estimates by method.** A scatterplot demonstrates the relationship between luciferase-based estimates of TS against MPRA-based estimates from each MPRA analysis method.
(TIF)

## Author Contributions

**Conceptualization:** Andrew R. Ghazi, Chad A. Shaw.

**Data curation:** Andrew R. Ghazi, Ed S. Chen.

**Formal analysis:** Andrew R. Ghazi.

**Funding acquisition:** Leonard C. Edelstein, Chad A. Shaw.

**Investigation:** Andrew R. Ghazi.

**Methodology:** Andrew R. Ghazi, Chad A. Shaw.

**Project administration:** Chad A. Shaw.

**Resources:** Leonard C. Edelstein, Chad A. Shaw.

**Software:** Andrew R. Ghazi, Ed S. Chen.

**Supervision:** Chad A. Shaw.

**Validation:** Andrew R. Ghazi, Xianguo Kong, Leonard C. Edelstein.

**Visualization:** Andrew R. Ghazi.

**Writing – original draft:** Andrew R. Ghazi, Chad A. Shaw.

**Writing – review & editing:** Andrew R. Ghazi, Chad A. Shaw.

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
