## [Decision Letter · Decision Letter 0]

22 Dec 2019

Dear Dr. Shaw,

Thank you very much for submitting your manuscript 'Bayesian modelling of high-throughput sequencing assays with malacoda' for review by PLOS Computational Biology. Your manuscript has been fully evaluated by the PLOS Computational Biology editorial team and in this case also by independent peer reviewers. The reviewers thought that the method could be potentially useful but raised some substantial concerns about the manuscript as it currently stands. In particular, the details of the method description and evaluation approach should be further clarified. The data used in the study should be made available. It would also be important to put this work in the context of existing literature and highlight the advantages and limitations. While your manuscript cannot be accepted in its present form, we are willing to consider a revised version in which the issues raised by the reviewers have been adequately addressed. We cannot, of course, promise publication at that time.

Sincerely,

Jian Ma

Associate Editor

PLOS Computational Biology

Thomas Lengauer

Methods Editor

PLOS Computational Biology

[LINK]

Reviewer's Responses to Questions

**Comments to the Authors:**

Reviewer #1: Summary:

This manuscript presents malacoda, a Bayesian method for identifying alleles with significantly different abilities to activate gene expression in an MPRA. malacoda accounts for variation due to unknown DNA concentrations and allows for users to incorporate prior information about variants, properties that existing methods do not have. In addition, unlike previous methods, it does not summarize the data before modeling. The manuscript reports results from simulations that show that malacoda is able to detect functional variants with higher accuracy than previous methods across multiple numbers of bar-codes per allele, array sizes, and fractions of functional variants. The manuscript also shows that malacoda tends to agree more with other methods on real MPRA data than the other methods agree with each other. In addition, the manuscript presents luciferase assays that malacoda’s results for sixteen variants, including multiple functional variants that were not identified as functional by other methods. Overall, I think that malacoda is a useful tool for using MPRA data to determine if non-coding variants are functional that is likely to be widely used because it is more accurate than existing methods and a thorough description of how it works as well as code needed to run it have been made publicly available.

Major Comments:

1. Figure 3A makes me concerned that malacoda’s precision is not very high because a large percentage of the simulated variants with transcription shifts greater than zero seem to be non-functional. Figure 3D enhances this concern because many of the luciferase estimates near zero seem to correspond to MPRA estimates that are further from zero. The manuscript would be more convincing if the authors could show that malacoda had high precision or at least substantially higher precision than other methods. In addition, since Figures 3A and 3D do suggest that malacoda does not report any non-functional variants to have transcription shift > 1.5, including the precision at different transcription shift cutoffs might also be helpful.

2. The paper mentions primary MPRA data, but this data is never described in detail. If the data is new to this paper, then a description of the data, including how the tested sequences were selected, what MPRA protocol was used, and how the plasmids were constructed, needs to be added. If the data was taken from another publication, then that publication needs to be cited.

3. The supplemental websites are extremely helpful. They provide many simulations that give readers intuition that is helpful in understanding the modeling decisions in this paper. They also help readers think about how much data would be needed to obtain good parameter estimates for the models in this paper. In addition, code is provided to generate the figures along with the packages needed to run the code, which enables readers to reproduce the figures and modify them in order to further improve their intuition.

Minor Comments:

Introduction:

1. The value of malacoda is not that it is Bayesian but that it does not have the drawbacks of previous methods – ignoring variation due to unknown DNA concentrations, summarizing the data prior to modeling, and not accounting for relevant prior information. I would re-structure the final paragraph of the introduction to emphasize its strengths over existing methods instead of its Bayesian nature.

Methods:

1. In line 144, the second dot should be a comma.

2. In lines 144-145, the reasons behind the definitions of the means for the negative binomial distribution were clear to me, but I am not sure if they would be clear to someone who has not thought about MPRAs before. It therefore might be useful to add a sentence explaining the reasons.

3. The definition of “95% highest density interval on TS” in line 171 was not clear to me; it would be great if this could be clarified.

4. The reason that the method for learning the gamma priors removes the need for post-hoc multiple testing correction, as claimed in line 193, was not clear to me. My understanding is that, even though all of the variants in the MPRA are used to learn the gamma priors, the effect of each variant on transcription is still tested separately, which would mean that multiple testing correction should still be required.

5. If an existing software package was used for model fitting, then the package and settings that were used should be added.

6. In the section on Simulation and Validation Studies, it would be great if a complete list the parameters used in the simulations could be provided (providing them as a supplemental table would be sufficient). If the supplemental websites contain every parameter used in the simulations, then directing the reader to the appropriate supplemental website should be sufficient.

7. It would be great if the authors could also compare area under the precision-recall curve because there are probably more non-functional variants than functional variants, so high sensitivity and specificity does not guarantee high precision.

8. In the section on Experimental Procedures, it would be helpful to add a description of the full experimental procedure for the new MPRA in this manuscript.

9. In the section on Experimental Procedures, it would be helpful to direct readers to the supplemental file containing the variants tested in luciferase assays.

Results:

1. At the beginning of the section on Biological Results, it would be helpful to include an explanation of why the number of luciferase reporter assays was not enough to overcome the amount of noise inherent to light intensity-based measurements or provide a citation of a paper that explains this.

2. Lines 294-296 state that the results from the luciferase assay are consistent with the MPRA estimates, but this seems to occur for only the largest transcription shifts. I think that this should be modified to say that the results are consistent for large transcription shifts.

3. Figure 4 suggests that evaluations were done using DeepSea-based priors, but there is no description of these results other than that the effect of rs11865131 was found using the DeepSea based priors. Figure 4B suggests that the results using the DeepSea-based priors were highly correlated with those without the priors. It would be helpful to add a summary of these results to the Results section.

4. The results provide evidence that rs11865131 is functional because it affects the activity of an intronic enhancer. The results about this variant would be more compelling if the authors added that this variant is not close to any exon-intron boundary, so it is unlikely to also affect splicing.

5. It would be helpful to add a description of how the analysis of rs11865131 overlap with ChIP-seq peaks was done (For example, were all peaks or only reproducible peaks used? When there were datasets from multiple labs for a TF or histone modification, which dataset was used?) and what the TFs and histone modifications are whose peaks overlap it.

6. If a luciferase assay was done for rs11865131, it would be helpful to add a description of the assay’s results in the Results section in addition to having them in a supplemental file.

Discussion:

1. The paper would be easier to follow if the description of how the models were fit was moved to the Methods section.

2. The phrase “seemingly worthwhile” in line 334 should be defined.

3. Line 342 mentions “an additional layer of hyper-parameters.” If these hyper-parameters were used to obtain the results in this paper, then a description of them should be added to the Methods section. If not, then it would be helpful to clarify that they are an option that the user can add but were not used to obtain this paper’s results.

4. It is not clear how much the DeepSea priors helped for the MPRAs described in this study. It would be helpful to add a section to the Discussion that provides some guidance to users about when priors are likely to be helpful.

Figures:

1. According to Figure 3A, the transcription shift for many non-functional variants is higher than the transcription shift for functional variants. It may appear to be this way because the difference between the density of points at the origin and on other parts of the y-axis is not viewable by eye. Modifying this figure to show the difference in density would be helpful. One option is to use smaller points. Alternatively, another part of the figure could be made that zooms in on the y-axis.

2. It would be helpful to add a panel to Figure 3 that illustrates how the results from malacoda are used to determine which variants are “functional.”

3. It would be helpful to add precision recall curves to Figure 3 to help the reader understand how frequently variants that are called positives are false positives.

4. It would be helpful to make the x- and y-axes the same scale in Figure 3D or add a line for y = x.

5. It would be helpful to add an explanation of the plots on the diagonals of the plot tables in Figure 4.

Supplement:

1. In the Description of point estimates of variance parameters section, it would be helpful to add a description of how the curve between maximum log-likelihood estimates of means and variances is fit.

2. In the Simulating variance parameter point estimates section, it would be helpful to add an explanation of why the chosen values are representative of the data from an MPRA or provide a citation of a paper that has such an explanation.

3. In the Simulating variance parameter point estimates section, it would be helpful to add an explanation of why the maximum likelihood estimates are systematically lower than the true value.

4. The NB likelihood surfaces section claims that the simulated draws described on the supplemental website “usually” allow for the true value. It was not clear to me why they do not always allow for the true value; in other words, I know that the true value will not always be drawn, but I am not sure why drawing the true value is not always possible.

5. In the Effect on mean estimation section, the dotted and orange lines on the x-axis were confusing to me. I would recommend removing them.

Reviewer #2: The authors presented a method and an R package called malacoda to model MPRA read count data with negative binomial distribution. The main contribution lies in modeling the mean and dispersion parameters as variables sampled from unknown gamma distribution. The paper is well written in general but there are some missing technical details. Major comments are below.

1. Authors should describe the Bayesian model in much more detail as it is now in the manuscript

a. One page 7, line 144 and 145, authors outlined the two generative NB models for DNA and RNA counts. Then they use Figure 2B to depict the hyperparameter of the gamma distributions that are used to model depath_s\\mu_{bc} and \\varphi_{DNA} for Counts_DNA. Similarly, another gamma distribution is used eto model depath_s\\mu_{bc}\\mu{allele} and \\varphi_{RNA} variables.

b. Just the generative model itself is not fully described in the main text. What’s distribution of \\mu_{bc}? What’s the distribution of \\varphi_{RNA}? If some of them are gamma distributed, what are the hyperparameters of these gamma distributions? Fixed or also estimated? Etc etc. No detail described here whatsoever.

c. The details on how the model is inferred is completely omitted. I understand the authors use out of the box optimizer Stan library to do the model fitting part. But that does not mean that there is no need to describe the necessary detail on the Bayesian inference.

d. In particular, given that \\mu_{bc} is unknown and is both DNA and RNA NegBin model, how is the coordinate ascent work in the variational Bayesian and how is it sampled in the no-U-turn Hamiltonian Monte Carlo sampling?

e. If variational Bayesian is used, what’s the proposed distribution?

f. If Hamiltonian Monte Carlo sampling, what’s the leap frog steps? What’s the step size? How the parameters are sampled? Together (not tractable)? Or separately in what order?

2. In Figure 3D, where authors used their in-house data to show malacoda, what’s the correlation across the 4 methods? It looks like malacoda is not better than the other 3 methods.

3. For rs11865131, in page 14 line 299, what’s the prediction scores from the other methods? Why is the variant not identified by those methods but by malacoda?

4. Page 14 line 306: How many such variants (i.e., rs11865131) exclusively identified by malacoda but not other methods? Can you count the number of ENCODE ChIP-seq peaks that each of the top 100 variants co-localize and compare them with the other 3 methods?

5. Authors says their model takes 50,000 MCMC samples, do the model converge in the end? Please show the plot of joint posterior or Hamiltonian energy (since authors is using HMC sampler) as a function of iterations.

6. Regarding more advanced prior, the authors should be aware that there are more advance supervised learning method that is train to predict MPRA signals using sequence features and epigenomic features. See this paper:

a. Li, Y., Shi, A., Tewhey, R., Sabeti, P., Ernst, J., & Kellis, M. (2017). Genome-wide regulatory model from MPRA data predicts functional regions, eQTLs, and GWAS hits. bioRxiv. http://doi.org/10.1101/110171

**Have all data underlying the figures and results presented in the manuscript been provided?**

Reviewer #1: Yes

Reviewer #2: Yes

PLOS authors have the option to publish the peer review history of their article (what does this mean?). If published, this will include your full peer review and any attached files.

Reviewer #1: No

Reviewer #2: No

---

## [Decision Letter · Decision Letter 1]

18 Mar 2020

Dear Dr. Shaw,

Thank you very much for submitting your manuscript "Bayesian modelling of high-throughput sequencing assays with malacoda" for consideration at PLOS Computational Biology.

As with all papers reviewed by the journal, your manuscript was reviewed by members of the editorial board and by several independent reviewers. In light of the reviews (below this email), we would like to invite the resubmission of a significantly-revised version that takes into account the reviewers' additional comments. In particular, one of the reviewers thought that the revision has not adequately addressed the concerns. 

We cannot make any decision about publication until we have seen the revised manuscript and your response to the reviewers' comments. Your revised manuscript is also likely to be sent to reviewers for further evaluation.

Sincerely,

Jian Ma

Deputy Editor

PLOS Computational Biology

Thomas Lengauer

Methods Editor

PLOS Computational Biology

Reviewer's Responses to Questions

**Comments to the Authors:**

Reviewer #1: I am impressed with the additional work that the authors have done to incorporate all of the reviewers’ feedback. The new version of the manuscript and the new appendix provide many helpful details that are necessary for fully understanding the presented computational and experimental work. In addition, the more comprehensive comparisons to previous methods and the new luciferase assay make me more convinced that malacoda is more effective than the previous methods described here at identifying alleles with differential ability to activate gene expression.

I have one new major comment:

1. I recently became aware of two relevant papers that the authors did night cite: Ashuach et al., Genome Biology, 2019 (https://genomebiology.biomedcentral.com/articles/10.1186/s13059-019-1787-z) and Kalita et al., Bioinformatics, 2018 (https://academic.oup.com/bioinformatics/article/34/5/787/4209990). It would be great if the authors could add an explicit explanation of the key advantages of malacoda over the methods described in these papers or the key differences between the types of problems that malacoda and these methods are capable of solving. If these methods are solving the same problem as malacoda, it would be great if the authors could add comparisons to these methods.

I have a few new minor comments:

Introduction:

1. In line 126, I would replace “wet bench” with “luciferase.”

Appendices:

1. The additional details in Appendix S1 are extremely informative for understanding exactly how malacoda works and how to run different parts of the method. malacoda seems to require the selection of multiple settings, including the number of warm-up samples, the number of samples per chain, and the total number of samples for MCMC. It would be great if the authors could add a description of how the recommended settings were selected.

2. Appendix S2 provides a helpful explanation of why having a prior on φ in the negative binomial distribution has the potential be beneficial, but it was clear what if any part of example 6 was illustrating malacoda’s prior on φ. It would be great if this could be clarified.

Reviewer #2: - It’d have been great if the authors were to describe what they have done to address my comments as opposed to just pointing me to a somewhere in the document. Especially, the changes are not highlighted relative to the original draft.

- For example, in response to my comment 1 on the distributions of the model, authors pointed me to line 174-179, where there is no distribution specified there but rather lines 151-156 have some added distributions. Authors point me to “Section 1 of the new S1 Appendix” on the rest of my comments expecting me to find the answers myself.

- For the variational Bayesian comments, the authors said “Stan’s variational interface with the R function rstan::vb(), …, former of these automatically transforms the parameters to the space of real numbers before using a Gaussian variational approximation.” How does the Gaussian approximation work on Negative Binomial? It may seems that my comments are too harsh. However, because this is a technical methodology paper, to me, the main contribution of this paper is on this detailed modeling.

- If the authors go through this round of the review, I would like to see in their response the model details instead of pointing me to somewhere else in the documents.

- For my comment 2, the correlation for malacoda is worse than the other two competitors (MPRAscore and even simple t-test). Authors said that this is not practically significant. But I thought the reason the authors show this scatter plot is to demonstrate that their method is better than other methods. Authors also said that “the assays shown were not selected to be representative of the most strongly functional variants”. But why not select the “representative of the most strongly functional variants” to test?

- On my comment 4, authors show a table comparing the consistency between top 100 variant predicted by each method with ChIP-seq peaks. It seems MPRAscore once again does better than malacoda. Even t-test has better lower false positive rate and comparable true positive rate. This does not seem to support malacoda as the method of choice.

- Comment 5 on 50,000 MCMC samples, instead of pointing to appendix, please add the plots in your response *if* the authors pass through this round.

**Have all data underlying the figures and results presented in the manuscript been provided?**

Reviewer #1: Yes

Reviewer #2: Yes

PLOS authors have the option to publish the peer review history of their article (what does this mean?). If published, this will include your full peer review and any attached files.

Reviewer #1: No

Reviewer #2: No
---

## [Decision Letter · Decision Letter 2]

10 May 2020

Dear Dr. Shaw,

Thank you very much for submitting your manuscript "Bayesian modelling of high-throughput sequencing assays with malacoda" for consideration at PLOS Computational Biology. As with all papers reviewed by the journal, your manuscript was reviewed by members of the editorial board and by several independent reviewers. Based on the reviewers' feedback, we are likely to accept this manuscript for publication, providing that you modify the manuscript according to the review recommendations.

Please prepare and submit your revised manuscript within 30 days by addressing to the additional comments from the reviewers. If you anticipate any delay, please let us know the expected resubmission date by replying to this email. 

Sincerely,

Jian Ma

Deputy Editor

PLOS Computational Biology

Thomas Lengauer

Methods Editor

PLOS Computational Biology

[LINK]

Reviewer's Responses to Questions

**Comments to the Authors:**

Reviewer #1: The changes that the authors have made to incorporate my and the other reviewers’ feedback have further improved the manuscript. I think that a major limitation of this manuscript is that there is little evaluation on real data other than in K562, an immortalized cancer cell line that is unlikely to be representative of any cell type in the human body. However, the ideal data for such evaluation does not exist, and the authors stated this limitation very clearly in the Discussion section, so I do not think that this limitation should hold back this manuscript from being published. I have come up with a few ideas that I have listed below that might help further demonstrate the value of malacoda relative to other methods. I also identified a few parts of the manuscript and github page that I would recommend further clarifying. Since the authors have made all of the code publicly available and provided details in their appendices about how to run malacoda, I think that malacoda could be come a widely used method for using MPRA data to identify putatively functional variants.

I have one new major comment:

1. I think that the paper would benefit tremendously from comparing malacoda to other methods on an additional real MPRA dataset. I recently became aware of a dataset that might be ideal for such a comparison: the dataset in Tewhey et al., Cell, 2016 (https://www.ncbi.nlm.nih.gov/pubmed/27259153). Since there is substantial eQTL data from lymphoblastoid cell lines from datasets like Geuvadis, I think that additionally showing that the putatively functional variants that malacoda detects in this dataset are more likely to overlap with eQTLs (or SNPs in linkage disequilibrium with eQTLs) than those detected by other methods would make the value of malacoda relative to other methods more apparent.

I have a few new minor comments:

Response to other reviewer:

1. The other reviewer brought up the important point that the putatively functional variants identified by MPRAscore are more likely to overlap ChIP-seq peaks than those identified by malacoda. The authors responded by comparing the DeepSEA DNase annotations between those variants across methods and evaluating the similarity between the annotations with a two-way ANOVA test. I found the description of the two-way ANOVA test a little confusing. My understanding is that the null hypothesis of the two-way ANOVA is that the means of the DeepSEA DNase scores are equal across methods. Thus, my understanding is that not rejecting the null hypothesis, does not imply that the null hypothesis is correct; it implies that the null hypothesis cannot be rejected. If the authors re-did the analysis on LCLs and found more overlap between putatively functional variants identified by malacoda than those identified by other methods, then I would not be as concerned about the ChIP-seq result. Alternatively, the authors could evaluate if the putatively functional variants identified by malacoda tend to be closer to TSS’s or closer to K562 DNase peak summits than those identified by other methods.

Introduction:

1. An exciting application of using MPRAs to identify putative functional variants is to help determine which of multiple variants in linkage disequilibrium that have all been associated with a disease are likely to be causal. I think that mentioning this near the beginning of the introduction might encourage more researchers to read the rest of the paper.

2. In lines 101-102, I think that “transcription binding” should be replaced with “transcription factor binding.”

3. In line 106, I would replace “has been unclear” with “has not been developed” (if that is accurate).

Methods:

1. In line 181, I would replace “learn” with “evaluate.”

Results:

1. The way that the p-values were computed was clear from the Appendix, but the way that the p-values were used to compute AUC and AUPR was not immediately intuitive to me. It would be helpful if this were described in more detail.

Discussion:

1. The Discussion section describes some extensions to malacoda that are available in the software package but are not described in detail or analyzed in this paper. I would recommend removing these and writing another paper about them that describes them in detail. Other researchers might be reluctant to use them if they do not have access to a clear explanation of how they work.

Figures:

1. The color-coding at the bottom of Figure 1 might be a little confusing to some readers because the red RNA could be interpreted as meaning that all of the RNA is coming from the first variant. I might instead make the RNA corresponding colors to the variant that produced it or make it similar to the brown color of the DNA that will get transcribed in the middle of the figure.

Appendices:

1. The code is generally easy to follow; removing the commented-out code would make some parts even easier to follow.

github page:

1. I think that a more detailed description of the inputs would make malacoda easier to use. Specifically, a description of the exact format of mpra_data would be helpful.

2. I had trouble finding the help documentation on the github. It would be great if a link to it could be added in a prominent location.

3. There seem to be some R scripts in the github that are not explained anywhere on the github. It would be great if a brief description of each R script could be added.

Reviewer #2: Authors have addressed the technical concern and clarity comments that I have. Please add legends to Figure 3B and 3C.

Also, Figure 3D median spear at zero is confusing. Authors should just add error bar to the bars displayed on the first three panels for median AUC, AUPR, non-zero correlation.

**Have all data underlying the figures and results presented in the manuscript been provided?**

Reviewer #1: Yes

Reviewer #2: Yes

PLOS authors have the option to publish the peer review history of their article (what does this mean?). If published, this will include your full peer review and any attached files.

Reviewer #1: No

Reviewer #2: No
---

## [Editor Report · Decision Letter 3]

9 Jun 2020

Dear Dr. Shaw,

We are pleased to inform you that your manuscript 'Bayesian modelling of high-throughput sequencing assays with malacoda' has been provisionally accepted for publication in PLOS Computational Biology.

Best regards,

Jian Ma

Deputy Editor

PLOS Computational Biology

Thomas Lengauer

Methods Editor

PLOS Computational Biology

---

## [Editor Report · Acceptance letter]

14 Jul 2020

PCOMPBIOL-D-19-01801R3 

Bayesian modelling of high-throughput sequencing assays with malacoda

Dear Dr Shaw,

I am pleased to inform you that your manuscript has been formally accepted for publication in PLOS Computational Biology. Your manuscript is now with our production department and you will be notified of the publication date in due course.

With kind regards,

Laura Mallard
